# In Vitro Hepatitis C Virus Infection and Hepatic Choline Metabolism

**DOI:** 10.3390/v12010108

**Published:** 2020-01-16

**Authors:** Kaelan Gobeil Odai, Conor O’Dwyer, Rineke Steenbergen, Tyler A. Shaw, Tyler M. Renner, Peyman Ghorbani, Mojgan Rezaaifar, Shauna Han, Marc-André Langlois, Angela M. Crawley, Rodney S. Russell, John P. Pezacki, D. Lorne Tyrrell, Morgan D. Fullerton

**Affiliations:** 1Department of Biochemistry, Microbiology and Immunology, Faculty of Medicine, University of Ottawa, Ottawa, ON K1H 8M5, Canada; kgobe057@uottawa.ca (K.G.O.); codwyer@uottawa.ca (C.O.); trenn061@uottawa.ca (T.M.R.); pghor093@uottawa.ca (P.G.); mreza007@uottawa.ca (M.R.); shan080@uottawa.ca (S.H.); langlois@uottawa.ca (M.-A.L.); acrawley@ohri.ca (A.M.C.); john.pezacki@uottawa.ca (J.P.P.); 2University of Ottawa Centre for Infection, Immunity and Inflammation and Centre for Catalysis Research and Innovation, Ottawa, ON K1H 8M5, Canada; 3Department of Medical Microbiology and Immunology and Li Ka Shing Institute of Virology, University of Alberta, Edmonton, AB T6G 2E1, Canada; rineke@ualberta.ca (R.S.); lorne.tyrrell@ualberta.ca (D.L.T.); 4Department of Chemistry and Biomolecular Sciences, Faculty of Science, University of Ottawa, Ottawa, ON K1N 6N5, Canada; tshaw@uottawa.ca; 5Chronic Disease Program, Ottawa Hospital Research Institute, Ottawa, ON K1H 8L6, Canada; 6Department of Medicine, Division of Infectious Diseases, The Ottawa Hospital, Ottawa, ON K1H 8L6, Canada; 7Department of Biology, Faculty of Science, Carleton University, Ottawa, ON K1S 5B6, Canada; 8Immunology and Infectious Diseases, Faculty of Medicine, Memorial University of Newfoundland, St. John’s, NL A1B 3V6, Canada; Rodney.Russell@med.mun.ca

**Keywords:** choline, hepatitis C virus (HCV), phosphatidylcholine, phospholipids, virus, Huh7.5, CTL1, SLC44A1, immunometabolism

## Abstract

Choline is an essential nutrient required for normal neuronal and muscular development, as well as homeostatic regulation of hepatic metabolism. In the liver, choline is incorporated into the main eukaryotic phospholipid, phosphatidylcholine (PC), and can enter one-carbon metabolism via mitochondrial oxidation. Hepatitis C virus (HCV) is a hepatotropic positive-strand RNA virus that similar to other positive-strand RNA viruses and can impact phospholipid metabolism. In the current study we sought to interrogate if HCV modulates markers of choline metabolism following in vitro infection, while subsequently assessing if the inhibition of choline uptake and metabolism upon concurrent HCV infection alters viral replication and infectivity. Additionally, we assessed whether these parameters were consistent between cells cultured in fetal bovine serum (FBS) or human serum (HS), conditions known to differentially affect in vitro HCV infection. We observed that choline transport in FBS- and HS-cultured Huh7.5 cells is facilitated by the intermediate affinity transporter, choline transporter-like family (CTL). HCV infection in FBS, but not HS-cultured cells diminished CTL1 transcript and protein expression at 24 h post-infection, which was associated with lower choline uptake and lower incorporation of choline into PC. No changes in other transporters were observed and at 96 h post-infection, all differences were normalized. Reciprocally, limiting the availability of choline for PC synthesis by use of a choline uptake inhibitor resulted in increased HCV replication at this early stage (24 h post-infection) in both FBS- and HS-cultured cells. Finally, in chronic infection (96 h post-infection), inhibiting choline uptake and metabolism significantly impaired the production of infectious virions. These results suggest that in addition to a known role of choline kinase, the transport of choline, potentially via CTL1, might also represent an important and regulated process during HCV infection.

## 1. Introduction

Hepatitis C virus (HCV) is a positive-strand (+)RNA virus that is inherently primate-borne and principally targets the liver [1]. HCV is part of the *Flaviviridae* family, which include viruses such as yellow fever virus, West Nile virus, dengue virus, and Zika virus [2]. Approximately 71 million people worldwide are infected with HCV, or approximately 1% of the global population [3]. Although some cases only lead to mild illness, most infected individuals (75–85%) develop chronic HCV infection. While recent advancements in direct-acting antiviral therapy have proven to be highly efficacious (~98% cure rates), barriers to therapy access, viral resistance, and low diagnosis rates minimizing efforts towards virus elimination, chronic HCV infection remains a significant medical concern. With no widely available vaccine, and culminating in chronic hepatitis, cirrhosis, as well as one of the world’s leading causes of death, hepatocellular carcinoma (HCC) [4], HCV infection has important implications in global health outcomes. Together with non-alcoholic fatty liver disease, HCV infection are the most common indications for a liver transplant in the Western world [3].

Crucial throughout all stages of development, choline plays an essential role in all tissues [5,6,7]. While choline is specifically taken into neuronal tissues via the high-affinity choline transporter (CHT1/*SLC5A7*), in non-neuronal tissues, low- and intermediate-affinity choline transport systems have been identified [8]. Organic cation transporters (OCTs/*SLC22A*) represent a family of low-affinity and promiscuous cation transporters. Intermediate-affinity transport is mediated by the choline transporter-like protein family (CTL1-5/*SLC44A1-5*), where CTL1 has been implicated as the main choline transporter in most tissues [9].

Phosphatidylcholine (PC) synthesis is a primary fate of hepatic choline, which accounts for approximately half of the phospholipid species in most mammalian cells [10]. Although in the liver and kidney, choline also serves as a methyl group donor via its mitochondrial oxidation to betaine [11]. In non-hepatic tissues, PC is exclusively made through the well-described cytidine diphosphate (CDP)-choline (or Kennedy) pathway; however, in the liver, PC supply is supplemented by the phosphatidylethanolamine (PE) *N*-methyltransferase (PEMT) pathway [12,13,14]. This pathway is thought to supply up to 30% of hepatic PC by methylating and converting PE to PC; however, it cannot fully compensate for disruptions in the CDP-choline pathway [15].

Previous studies have demonstrated that (+)RNA viruses impact cellular phospholipid metabolism [16,17], and certain viruses have been observed to increase the PC content in distinct cellular membranes during sustained infection [18]. Interestingly, the role of choline kinase alpha (CHKα), the first step of the CDP-choline pathway, has proven to be a requisite component in maintaining the integrity of the HCV membranous web, the site of viral replication located on the ER [19,20]. Moreover, CHKα was shown to be essential in shuttling the phosphatidylinositol-4-kinase IIIα-nonstructural protein 5A HCV assembly complex to the viral replication complex, giving further credence that PC metabolism may be integral to the HCV life cycle [21]. Finally, PC forms the principle membrane component of hepatocyte-derived ApoB-containing lipoproteins such as very low-density lipoproteins (VLDL). It has been previously shown that HCV subverts VLDL to make lipoviral particles that subsequently undergo a multi-step endocytosis that aides in HCV infection of hepatocytes [22,23,24,25]. Interestingly, it has recently been shown that Huh7.5 cells, which represent one of the most well-characterized cell culture models for studying hepatic viral biology, behave very differently when cultured in human serum (HS) compared to normal fetal bovine serum (FBS) [25,26]. HS-cultured Huh7.5 cells are more polarized, express more functional hepatocyte markers, and dramatically increase VLDL secretion. Although some aspects of choline metabolism have been investigated in sustained infection models, the importance of choline transport and the cellular response of PC metabolism to the HCV infection have yet to be fully described.

Therefore, in this study we sought to interrogate this potential link and observed that the CTL family of transporters mediated the functional uptake of choline in Huh7.5 cells. Concordant with decreases in choline uptake and CTL1 expression, the rate of flux through the CDP-choline pathway was lower after 24 h of HCV infection in FBS-cultured cells; however, no differences were observed in HS-cultured cells or cells infected for 96 h. The inhibition of choline transport during early infection in both FBS- and HS-cultured cells augments viral replication, an effect that was lost or diluted by 96 h. Finally, diminished choline metabolism dramatically lowers HCV infectivity index in both culture systems. Our results suggest that choline transport represents a previously underappreciated aspect of PC metabolism that is both able to regulate and be regulated by HCV infection in FBS- and HS-cultured Huh7.5 cells.

## 2. Materials and Methods

### 2.1. Huh7.5 Cell Culture

Huh7.5 cells were a kind gift from C.M. Rice (Rockefeller University). FBS-cultured cells were cultured in Dulbecco’s modified Eagle medium (DMEM) containing 10% FBS (Wisent, Saint-Jean-Baptiste, QC, Canada) and 1% penicillin/streptomycin (Thermo Fisher Scientific, Ottawa, ON, Canada ). Cells were maintained under 100% confluency and were split using standard trypsin/EDTA protocol. Cells of different passages on different days were considered as biological replicates. HS-cultured cells were grown as previously described [25], where 2% HS (Valley Biomedical, Winchester, VA, USA) was used in place of FBS. These cells form confluent layers of growth arrested cells, that do not need further sub-culturing.

### 2.2. JFH-1 HCV Infection Protocol

The initial harvest of JFH-1 HCV was following electroporation into FBS-cultured cells, as previously described [27]. Viral production (RNA/mL and 50% tissue culture infectious dose (TCID50/mL) was further monitored. Supernatants were collected after 4 days and these viral stocks were either used to propagate more FBS-cultured virus or for HS-cultured infection experiments described below. For FBS-cultured cells, Huh7.5 cells were plated at 70% confluency and infected with live JFH-1 strain HCV virions at a MOI = 1 for 4 h containing 5% lipoprotein deficient serum. Fully differentiated and confluent HS cultured cells were infected at an MOI = 1. Following this 4 h incubation, the media was removed and cells were washed three times before being cultured in FBS or HS media. Experiments were performed 24 h after infection for FBS cells and 96 h after infection for HS cells.

### 2.3. JFH-1 HCV Titration

Infectious virus particles were quantified using a focus forming assay. Virus supernatant was serially diluted 10-fold in serum-free medium and dilutions were used to infect Huh7.5 cells seeded (at 5 × 10^4^ per well) onto 8-well Lab-Tek II chamber slides (NUNC) for 4 h. Following incubation, the infectious medium was removed and replaced with fresh medium containing 10% FBS and 1X NEAA. Forty-eight hours post infection, cells were washed with 1X PBS and fixed with acetone for 5 min before staining with HCV core monoclonal antibody (1:100; Thermo Fisher Scientific, Ottawa, ON, Canada; MA1080), followed by a secondary antibody, Alexa Fluor 488-conjugated goat anti-mouse (1:250; Thermo Fisher Scientific, Ottawa, ON, Canada; A-11029). Viral titers are expressed as the number of focus-forming units (FFU) per mL of supernatant.

### 2.4. [^3^H]-Choline Uptake Experiments

The uptake of choline in Huh7.5 cells was determined by using [^3^H]-choline (Perkin Elmer, Woodbridge, ON, Canada) to evaluate uptake the desired time points. Prior to the addition of radiolabeled choline, Huh7.5 cells seeded in 24-well plates were washed twice with PBS then incubated with Krebs-Ringer-HEPES buffer (KRH; 130 mM NaCl, 1.3 mM KCl, 2.2 mM CaCl_2_, 1.2 mM MgSO_4_, 1.2 mM KH_2_PO_4_, 10 mM HEPES pH 7.4, 10 mM glucose) for 1 h at 37 °C prior to treatment to remove extracellular choline. Following KRH incubation, [^3^H]-choline (1 µCi/mL in KRH) was added to the cells for the desired time points (0–30 min) and incubated at 37 °C. For uptake kinetics, cells were incubated in increasing concentrations of non-radiolabeled choline for 10 min. The [^3^H]-choline was then removed, and the cells were washed twice with KRH buffer, then lysed in 300 µL of 0.1 M NaOH. Cells were flash frozen using liquid nitrogen, thawed then the lysate was scraped and collected. Cell lysate was centrifuged at 20,000× *g* for 5 min and supernatant was collected. Cellular protein amounts were determined by BCA protein quantification assay according to manufacturer’s instructions (Thermo Fisher Scientific, Ottawa, ON, Canada) and radioactivity was quantified through liquid scintillation counting (LSC). Choline uptake and kinetics were calculated as previously described [28].

### 2.5. [^3^H]-Choline Incorporation into PC

Huh7.5 cells were seeded in 12-well plates and incubated at 37 °C up to 8 h with DMEM containing 0.5 µCi/mL of [^3^H]-choline. Cells were then washed twice with PBS and 250 µL of PBS was the added to the cells, which were then flash frozen in liquid nitrogen. Following the collection of cell lysate, of which a 50 µL aliquot was counted by LSC, we performed a total lipid extraction protocol described by Bligh and Dyer [29]. The organic phase was evaporated under nitrogen and resuspended in 25 µL of chloroform to concentrate the lipid species. The concentrated solution was then subjected to thin layer chromatography as previously described and radioactivity corresponding to PC was counted by LSC [30].

### 2.6. Choline Inhibition by HC3

FBS-cultured Huh7.5 cells were infected as described above in the presence or absence of 20 or 200 μM HC3 for 24 h. Following treatment, cells were either processed for RNA or labeled with [^3^H]-choline as described above. HS-cultured Huh7.5 cells were infected as described above and treated with 20 or 200 μM HC3 at 72 h post-infection, for a total of 24 h before being harvested for RNA quantification.

### 2.7. RNA Isolation and Quantification

Total RNA was extracted using the TriPure reagent protocol (Roche Life Sciences, Mississauga, ON, Canada). Following extraction, the RNA was resuspended in 20 µL of RNAse/DNAse-free H_2_O (Wisent) and concentration equalized. RNA was then reversed transcribed using the QuantiNova™ kit (Qiagen, Toronto, ON, Canada) according to the kit’s instructions. Transcript expression was determined by using primers obtained from PrimerBank, designed through NCBI PrimerBlast, or from previous published studies [31] (Appendix A). These primers were used in conjunction with the BrightGreen 2× qPCR mastermix (ABM, Vancouver, BC, Canada). The relative transcript expression was determined using the delta-delta *Ct* method [32] and normalized to the averages of *β ACTIN* and *HSP-90*. The efficiencies of all primers were validated before use. For quantification of viral RNA, infected cells underwent total RNA extraction as described above. Reverse transcription was initiated using the OneScript cDNA Synthesis Kit (ABM, Vancouver, BC, Canada) as per the kit instructions, using random hexamer primers. Viral RNA was assessed using previously published primers (Appendix A) and shown relative to *HSP-90* and/or *β ACTIN*. TCID50s were determined as described [33].

### 2.8. Immunoblotting

To investigate protein expression, cells were lysed in native lysis buffer (50 mM Tris-HCl pH 7.4, 150 mM NaCl 1 mM EDTA, 100 μM sodium orthovanadate, and protease inhibitor cocktail tablet; Roche Life Sciences, Mississauga, ON, Canada). Lysate was equalized and loaded onto an 8% denaturing SDS-PAGE gel. Following electrophoresis, duplicate gels were transferred using the Trans-Blot^®^ system (BioRad, Mississauga, ON, Canada) onto PVDF membranes (1.3 A for 17 min) using Bjerrum Schafer-Nielsen buffer (48mM Tris, 39 mM glycine, 20% methanol). The membranes were blocked in 5% BSA in TBS-T (20 mM Tris, 150 mM NaCl, 0.05% Tween^®^ 20) for 1 h then incubated overnight at 4 °C with primary rabbit antibody targeted towards CTL1 (C-terminal antibody previously described [34]), CTL2 (Abcam #177877) or GAPDH (CST 8884S). Following overnight incubation, the membranes were washed 4 times with TBS-T and then incubated for 1 h with a 1:1000 dilution of HRP-conjugated anti-rabbit secondary antibody made in 5% BSA in TBS-T (GAPDH was HRP-conjugated, so was not exposed to secondary). The membranes were then washed four times with TBS-T then treated with Clarity™ Western ECL solution (BioRad, Mississauga, ON, Canada) according to manufacturer instructions. Membranes were visualized and imaged using LAS 4010 ImageQuant imaging system (GE, Mississauga, ON, Canada).

### 2.9. Statistics

All statistical analyses were performed using Prism7 (GraphPad Software Inc., San Diego, CA, USA). Experiments consisting of only two groups were assessed by Student’s *t*-test. Choline uptake kinetic measurements were fit to Michaelis-Menten equations and uptake inhibition curve was fit to a logarithmic (inhibitor) vs. response curve fit. Experiments involving two groups and multiple treatments were compared using two-way ANOVA with a Tukey post-hoc analysis. For viral replication experiments, statistical significance was determined by a one-way ANOVA within each condition (FBS-control, FBS-HCV, HS-control and HS-HCV), with a Tukey’s multiple comparisons test. All data represent mean ± SEM, unless specified in the figure legend.

## 3. Results

### 3.1. Determination of Choline Transport Kinetics

To establish a kinetic profile of choline uptake in the Huh7.5 cell line cultured in FBS, we first performed uptake experiments using [^3^H]-choline and established that transport was linear over the course of 30 min. Following this, we determined that the apparent affinity for choline (*K*_M_) was 66.8 ± 9.08 µM and that the *V*_max_ was 296.3 ± 14.45 µmol/min (Figure 1A), indicative of an intermediate-affinity transport system [8,9]. There is evidence that hepatic choline transport is mediated by the CTL family, namely CTL1, as well as potentially the low-affinity OCT family [8,9,35]. We next used hemicholinium-3 (HC-3), a potent pharmacological inhibitor of high and intermediate affinity choline transporters, which revealed that ~90% of choline transport was sensitive to HC-3 uptake inhibition (Figure 1B). In addition to HC-3, we also aimed to assess the potential contribution of the OCTs by using a well-characterized OCT inhibitor, quinine. While we preformed a dose-response curve (data not shown), at the highest dose (200 μM), choline uptake remained unchanged by OCT inhibition (Figure 1C).

We next profiled the transcript expression of the relevant choline transporters in FBS-cultured Huh7.5 cells. The relative expression of CTL1 (*SLC44A1*) transcript was higher compared to that of other family members, while only OCT1 and OCT2 (*SLC22A1* and *SLC22A2*) were detectable from the OCT family (Appendix A). Additionally, to rule out the typically neuronal high affinity choline transporter CHT1/SLC5A7 as a contributor in FBS-cultured Huh7.5 cells, we confirmed that expression levels were undetectable.

There have been important distinctions observed between Huh7.5 cell cultured FBS as compared to those cultured in HS. We sought to take advantage of these related but discordant culture systems. Importantly, the choline uptake kinetics in HS-cultured cells was very similar to FBS-culture conditions (Appendix A), indicating that intermediate-affinity (CTL-mediate) transporters were likely dominant.

### 3.2. Effects of In Vitro HCV Infection on Choline Transporter Expression and Transport

To investigate the potential role of choline and PC metabolism in relation to HCV pathogenicity, we next aimed to determine how choline transporter gene expression might be affected by in vitro HCV, 24-h, and 96-h post-infection. First, we compared non-infected, naïve FBS- and HS-cultured Huh7.5 cells, to those infected with the JFH-1 strain of HCV (MOI = 1). While there were inherent differences in the transcript expression profile between FBS- and HS-cultured cells, the only difference between HCV infection and naïve cells was a decrease in the expression of *SLC44A1* (gene encoding CTL1) at 24 h in FBS-cultured cells (Figure 2A). However, this decrease was recovered by 96 h post-HCV infection (Figure 2B).

From parallel treated cells, we measured the protein expression of the two main choline transporter-like family members, CTL1 and CTL2. We did not investigate the protein content of OCT1 given the lack of contribution of the OCTs to choline transport in Huh7.5 cells (Figure 1). Similar to transcript expression, there was a significant decrease in the total protein content of CTL1 in 24 h HCV-infected FBS-cultured cells compared to uninfected controls (Figure 3A). The decreased content of CTL1 was not observed at 96 h post-HCV infection (Figure 3B). Moreover, similar to transcript levels, the total levels of CTL1 and CTL2 proteins were different between FBS- and HS-cultured cells, independent of HCV infection, where both CTL1 and CTL2 levels were consistently lower in HS-cultured cells compared to those grown in FBS (Figure 3).

Given that the only perturbation to choline transporter expression was observed in FBS-cultured cells at an early (24 h) time point, we next interrogated whether these changes in CTL1 transcript and protein expression were associated with differences in choline transport and metabolism. After 24 h, FBS-cultured infected cells displayed a small but significant impairment in choline uptake and a 2-fold reduction in *V*_max_ (Figure 4A,B), with no change in apparent affinity for choline (*K*_M_). Accompanying this, 24 h HCV infection lowered [^3^H]-choline incorporation into PC compared to non-infected control cells (Figure 4C).

These data suggest that while baseline differences in choline transporter expression exist between Huh7.5 cells cultured in FBS and HS, that the only significant effect of HCV infection is observed in FBS-cultured cells and only at an early time point during infection.

### 3.3. Inhibiting Choline Uptake Alters Early HCV Replication

We next aimed to ascertain if inhibiting choline availability would affect stages of HCV infection. Using HC-3 as a pharmacological means of achieving choline transport inhibition, FBS- and HS-cultured Huh7.5 cells were infected with HCV as above (MOI = 1), but in the presence or absence of 20 or 200 μM HC-3 for 24 h. As above (Figure 4), in the absence of HC-3, 24 h HCV infection resulted in a diminished incorporation of [^3^H]-choline into PC (Appendix A). Treatment with 20 μM HC-3 reduced incorporation into PC in uninfected cells, whereas 200 μM treatment resulted in a dramatic reduction in choline incorporation into PC independent of HCV infection (Appendix A), validating the inhibition of choline uptake.

To investigate early (24 h) and more chronic (96 h) stages of HCV infection in the context of viral replication and virion infectivity, we aimed to maintain a consistent and relatively acute window of choline transport inhibition. For the acute (24 h) infection, cells were treated concurrently with vehicle (DMSO), 20 or 200 μM HC-3, resulting in a 24 h period of infection and choline inhibition. These cells were harvested after 24 h. For the more chronic (96 h) infection, cells were infected but only treated with and without HC-3 after 72 h, resulting in a 96 h HCV infection with a 24 h inhibition of choline uptake. As an indication of viral replication, we determined the relative amount of HCV RNA in control and HCV-infected cells. Interestingly, in the presence of HC-3, the relative amount of HCV in FBS-cultured cells after 24 h was dose-dependently higher than that of control HCV-infected FBS-cultured cells (Figure 5A). Moreover, while the relative amount of HCV was consistently increased in HS-cultured cells compared to the FBS-culture system (as has previously been shown [25]) there remained a stimulatory effect of choline uptake inhibition (200 μM) in the HS-cultured cells (Figure 5A) at the early 24 h time point. In cells infected with HCV for 96 h, but treated with HC-3 for the last 24 h, no differences in viral replication (relative viral RNA) were observed in FBS-cultured cells, while a residual, but significant effect was seen in HS-cultured cells treated with 200 μM HC-3 (Figure 5B). Therefore, limiting the availability of choline was associated with an increase in viral replication early during infection in both FBS-and HS-cultured Huh7.5 cells, but only in HS-cultured cells in a more chronic in vitro infection model.

### 3.4. Limiting Choline Availability during HCV Infection Inhibits Viral Infectivity

While limiting choline availability augmented the relative amount of intracellular viral RNA during early infection (to a lesser extent during more chronic infection), we aimed to determine how this might affect the infectious nature of the virus. Given that 24 h is not a sufficient amount of time for HCV to infect, replicate, and undergo shedding, we did not assess the effect of choline uptake inhibition after 24 h. When FBS- and HS-cultured cells were infected and treated with HC-3 at 72 h post-infection for a further 24 h, the HCV virus secreted from HS-cultured cells treated with the choline uptake inhibitor were significantly less infectious (Figure 6). As previously demonstrated, virus from vehicle-treated HS-cultured cells was significantly more infectious when compared to virus taken from FBS-cultured cells. Moreover, virus from FBS-cultured cells showed no significant difference in infectivity, although a modest downward trend was observed dose-dependently (Figure 6).

## 4. Discussion

The human Huh7 and related cell lines have long been used to address key biological questions with regards to HCV life cycle. While typical culture conditions include the use of standard 10% FBS, it has recently been demonstrated that culturing of Huh7.5 cells in media supplemented with HS, dramatically shifts the cellular and metabolic phenotype of the cells [25,26]. We sought to take advantage of this model system to interrogate the role of HCV infection on the expression of choline transporters, as well as whether inhibition of choline uptake altered viral replication and release.

Hepatic choline can be used for phospholipid synthesis or can enter the one-carbon pathway via its oxidation to betaine. While numerous studies have focused on hepatic choline metabolism, few have looked to address the initial cellular uptake of choline. Here, we provide evidence that in Huh7.5 cells, two choline transporter families are expressed, the low-affinity broad spectrum OCT family and the intermediate-affinity CTL family. However, on a functional level, when [^3^H]-choline uptake measurements were fit into a Michaelis-Menten model, it was suggestive of an intermediate affinity for choline (*K*_M_ of 66.8 ± 9.1 µM) thus pointing to the CTL transporters [36]. Moreover, when the contribution of both CTL and OCT transporters was assessed through selective pharmacological inhibition it was clear that only HC-3-sensitive (CTL-mediated) transport is present in Huh7.5 cells [34,35]. Interestingly, while OCT family members seemed to play a minor role in choline uptake, the relevance to human hepatocytes in vivo warrants further investigation [37,38].

HCV infection leads to widespread hijacking of the host-cell machinery thereby dysregulating a myriad of pathways to allow for its sustained chronic infection of the host [39,40,41,42,43]. Having an innate dependence on enzymes involved in lipid pathways at every stage of its replicative cycle, the link between HCV and lipid metabolism has been well established [24,44,45]. Our study aimed to address another aspect of lipid metabolism in HCV life cycle, the uptake of choline and its incorporation into PC.

Infection with (+)strand RNA viruses, such as HCV, has been shown to cause an increase in PC amounts at the site of viral replication [16]. Conversely, it was demonstrated that upon HCV infection, the total amount of PC was reduced relative to non-infected cells [46]. The latter is entirely consistent with the lower incorporation of choline into PC we observe 24 h post-infection in FBS-cultured cells (the same culture system used by [46]). Interestingly, HCV requires an interaction with CHKα, the initial enzyme responsible for the phosphorylation of choline in the CDP-choline pathway [46] and there is evidence that CHKα activity, but not its PC-synthesizing function is critical.

The biological significance of lower PC levels in HCV-infected cells remains unclear. In vivo, the CDP-choline pathway accounts for approximately 70% of PC, whereas the remaining levels are produced via PEMT and the methylation of PE. Upon HCV infection, there is a reduction in PEMT transcript and protein expression [46]. Our results add to this narrative by demonstrating a reduction in the choline transporter CLT1 after 24 h of HCV infection (Figure 2 and Figure 3), which was associated with a decrease in choline uptake and flux through the CDP-choline pathway (Figure 4). Together, this may explain the lower levels of PC. However, these results only provide a snapshot of what is happening at 24 h post-infection in vitro. It will be important to consider not only the content, but the fatty acid composition of PC species, which also change with HCV infection [47].

Given that catalytic inhibition of CHKα and not necessarily flux through the CDP-choline pathway has been tied to viral replication [46], it was somewhat surprising that the inhibition of choline uptake augmented viral RNA replication after 24 h. Inhibiting PC synthesis via blockade of the rate limiting enzyme in the CDP-choline pathway, CCTα, was not shown to effect HCV replication [16,21,46]. One potential explanation for this result and discrepancy might be timing. We chose to identify changes to host cell metabolism and viral replication after 24 h, where previous work has only looked at 72 h and beyond. Interestingly, when we investigated relative amounts of viral RNA at 96 h post-infection, inhibiting choline uptake and metabolism during the last 24 h had only a slight, but significant effect in HS-cultured cells. While the reason for the discrepancy between time points and culture conditions remains unclear, future work should aim to test genetic knockdown of choline transporters in this context, or look to inhibit chronically with HC-3. It might also be that limiting choline uptake and availability drives up the expression and activity of other PC-generating pathways (PEMT and PC degradation via phospholipases) to scavenge the PC necessary for viral replication. Interestingly, phospholipase D, which cleaves existing PC to yield free choline and phosphatidic acid, is upregulated in an in vitro model of HCV [48].

HS-cultured cells, unlike FBS-cultured cells displayed no differences in transporter expression upon HCV infection. While this cell model may represent a system that is more physiologically similar to fresh primary human hepatocytes, there are potential explanations for the divergent observations. The metabolic demands of the cells differ dramatically since FBS-cultured cells divide readily and HS-cultured cells do not. This calls into question whether HCV differentially competes for cellular nutrients such as choline in proliferative vs. quiescent cells.

It remains unclear as to why there is an apparent divergence between relative amounts of viral RNA (Figure 5) and the measure of infectious viral particles (TCID50; Figure 6). Future work could address limitations of the current experimental work flow and should test whether choline uptake inhibition during the first 24 h of infection (when relative viral RNA was dramatically increased), then correlates to altered number of infectious particles at 96 h. However, given that TCID50 measurements were conducted on cells that had been infected for 96 h, we can conclude that the inhibition of choline uptake during this period of infection results in slight but significant increases in viral RNA (Figure 5B), but does not lead to—rather, dramatically inhibits—viral infectivity. Future work is needed to tease out whether this apparent loss of infectious properties is due to a lack of choline transport, changes in CHKα activity, PC content/composition, and/or a combination. Importantly, there may also be off-target effects with the use of any pharmacological inhibitor that will only be ruled out with the use of genetic knockdown in future studies.

In addition to possible timing issues, another consideration is our strict focus on PC metabolism. Choline, once transported into the hepatocyte, can enter into the mitochondria and become oxidized to betaine as the first committed step toward its incorporation as a methyl group donor (for DNA and protein methylation). Along these lines, an initial characterization of hepatic choline metabolic fate has been reported in primary rat vs. a documented HCC cell line, which highlighted the alternative fates of choline toward each arm of the phospholipid and methylation pathways [35]. It will be important for future work to consider the full metabolic consequence of altered choline homeostasis during all stages of in vitro HCV infection.

## 5. Conclusions

In conclusion, we demonstrate that short term HCV infection of FBS-, but not HS-cultured Huh7.5 cells, results in a substantial impairment of choline uptake and incorporation into PC. This is associated with a lower expression of the choline transporter CTL1. We also show that in both culture systems, when choline uptake is inhibited, this paradoxically potentiates relative viral RNA amount at an early, but not later, time point. Finally, in chronically HS-cultured Huh7.5 cells, HCV infection had no effect on choline transporter expression; however, choline uptake inhibition significantly impaired the infectious nature of the virus particles. Taken together, these results demonstrate the potential importance of choline metabolism in HCV and potentially other hepatic viral infections.

## Figures and Tables

**Figure 1 viruses-12-00108-f001:**
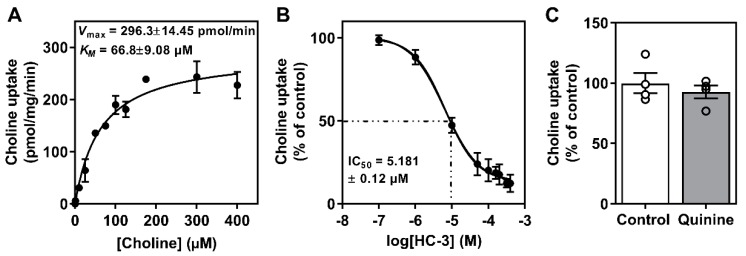
Characterization of Huh7.5 cell choline uptake. (**A**) Choline uptake saturation kinetics fit to a Michaelis-Menten curve. (**B**) Inhibition of choline uptake in response to hemicholinium-3 (HC3). (**C**) Choline uptake inhibition in response to 200 μM quinine (OCT inhibitor). Data are mean ± SEM and represent 3–4 independent experiments.

**Figure 2 viruses-12-00108-f002:**
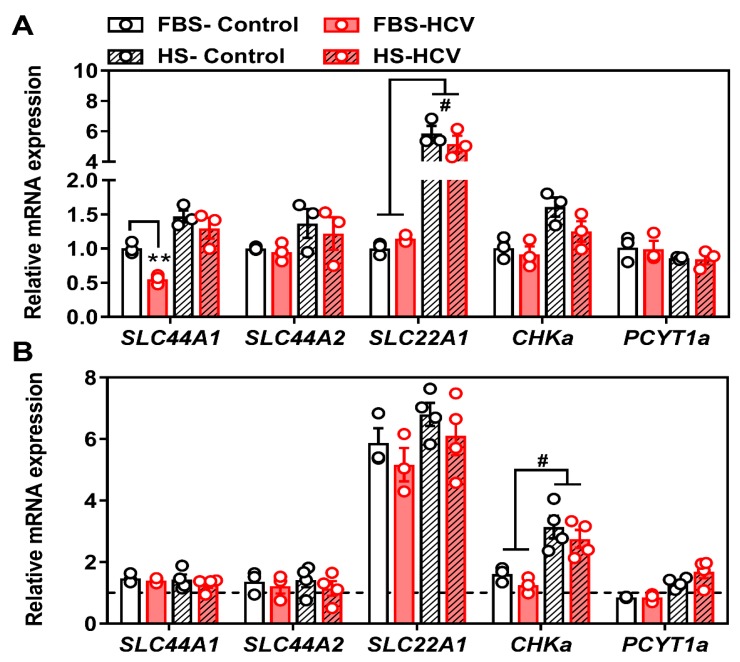
Choline transporter mRNA expression at 24 and 96 h post-HCV infection. FBS- and HS-cultured Huh7.5 cells were infected at an MOI of 1 for (**A**) 24 or (**B**) 96 h before cells were washed and RNA isolated to measure choline transporter (*SLC44A1*; choline transporter-like 1, SLC44A2; choline transporter-like 2 and SLC22A1; organic cation transporter 1) as well as CDP-choline pathway (*CHKa*; choline kinase alpha and *PCYT1a*; phosphocholine cytidylyltransferase alpha) transcript expression. All groups are shown relative to uninfected FBS-cultured control cells at 24 h and normalized to the average of *β ACTIN* and *HSP90*. The hashed line in (B) represents the level of uninfected FBS-cultured control cells at 24 h. Data are mean ± SEM and represent three independent experiments. Statistical significance was determined by two-way ANOVA with a Tukey post-hoc analysis (within each transcript), such that ** represents *p* < 0.01 compared to uninfected control within serum culture condition and # represents *p* < 0.05 compared between FBS- and HS-cultured conditions.

**Figure 3 viruses-12-00108-f003:**
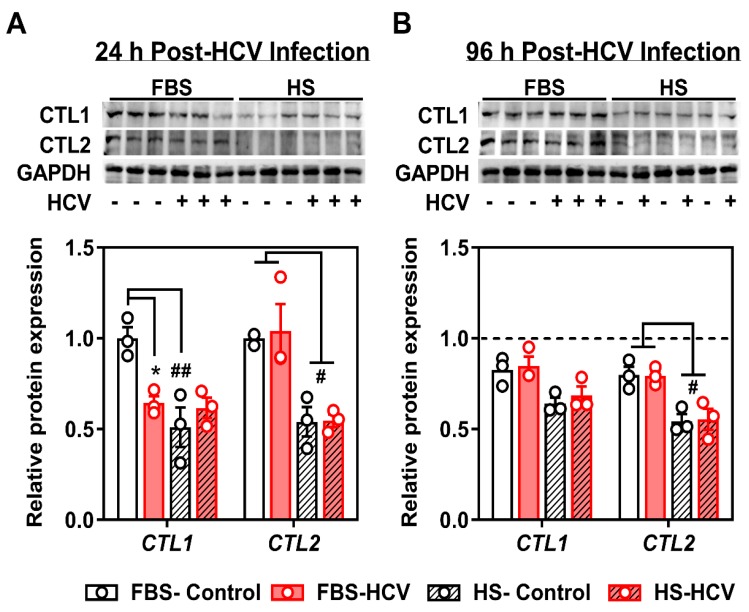
Choline transporter protein expression at 24 h and 96 h post-HCV infection. FBS- and HS-cultured Huh7.5 cells were infected at an MOI of 1 for (**A**) 24 h or (**B**) 96 h before cells were washed and protein isolated to measure choline transporter (CTL1; choline transporter-like 1 and CTL2; choline transporter-like 2) total protein expression. Samples were run on two individual gels, but transferred to a common membrane. Densitometry analyses depicts the density of CTL1 and CTL2 normalized to GAPDH and shown relative to uninfected FBS-cultured control cells at 24 h. The hashed line in (**B**) represents the level of uninfected FBS-cultured control cells at 24 h. Data are mean ± SEM and represent three independent experiments. Statistical significance was determined by two-way ANOVA with a Tukey post-hoc analysis (within each protein), such that * represents *p* < 0.05 compared to uninfected control within serum culture condition, and # and ## represents *p* < 0.05 and *p* < 0.01 compared between FBS- and HS-cultured conditions, respectively.

**Figure 4 viruses-12-00108-f004:**
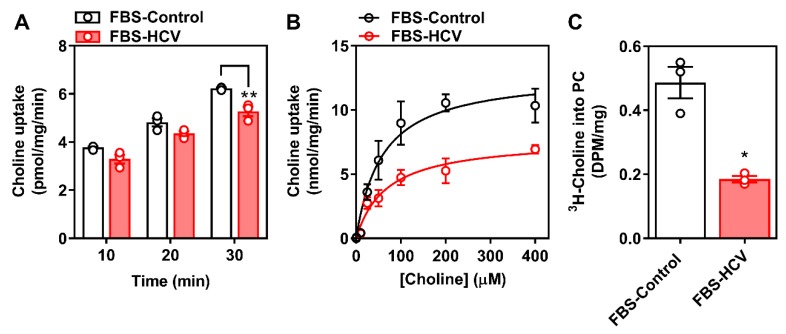
Choline uptake and incorporation into PC 24 h post-HCV infection. FBS-cultured Huh7.5 cells were infected at an MOI of 1 for 24 h. (**A**) [^3^H]-choline uptake was determined over the course of 30 min; (**B**) [^3^H]-choline uptake saturation kinetics were determined in the presence of increasing concentration of non-radiolabeled choline and fit to a Michaelis-Menten curve; and (**C**) [^3^H]-choline incorporation into PC was then determined. Data are mean ± SEM, are representative of three independent experiments and normalized to total protein content, where * and ** represent statistical significance compared to uninfected control cells at *p* < 0.05 and *p* < 0.01, respectively.

**Figure 5 viruses-12-00108-f005:**
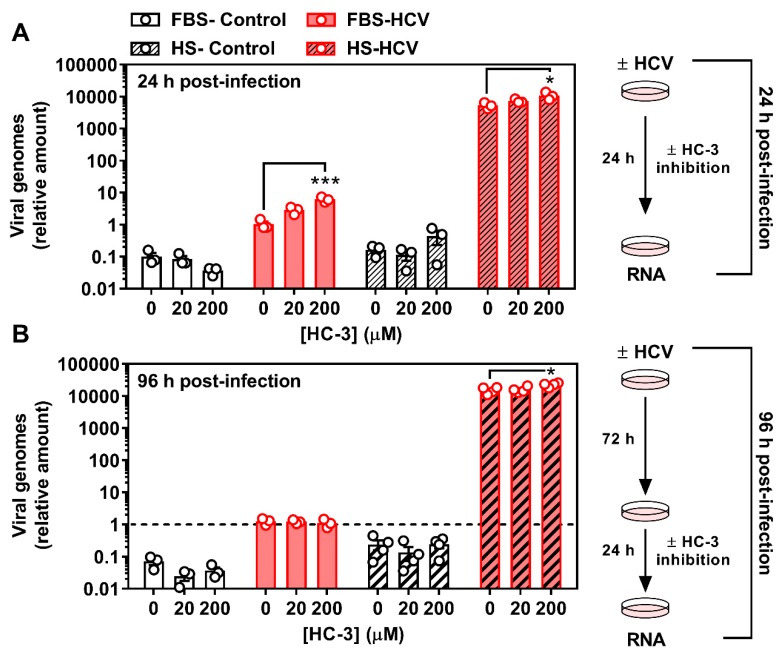
Inhibition of choline uptake augments HCV replication after 24 h. FBS- and HS-cultured Huh7.5 cells were infected at an MOI of 1 for (**A**) 24 h in the presence or absence of 20 or 200 μM hemicholinium-3 (HC-3) to inhibit choline uptake and (**B**) 96 h in the presence or absence of 20 or 200 μM HC-3 for the last 24 h of infection. Cells were washed and RNA was isolated to assess HCV RNA expression as an indication of viral replication. All treatments are shown relative to infected vehicle FBS-control cells and normalized to the average of *β ACTIN* and *HSP90*. Data are mean±SEM and are represent of 3-4 independent experiments. The hashed line in (**B**) represents the level of infected FBS-cultured control cells at 24 h. Statistical significance was determined within each culture system and HCV group, such that 0, 20 and 200 μM HC-3 from FBS- and HS-cultured cells were each determined by one-way ANOVA where * and *** represent *p* < 0.05 and *p* < 0.001, respectively (determined by a Tukey *posthoc* test) compared to vehicle control (0 μM HC-3).

**Figure 6 viruses-12-00108-f006:**
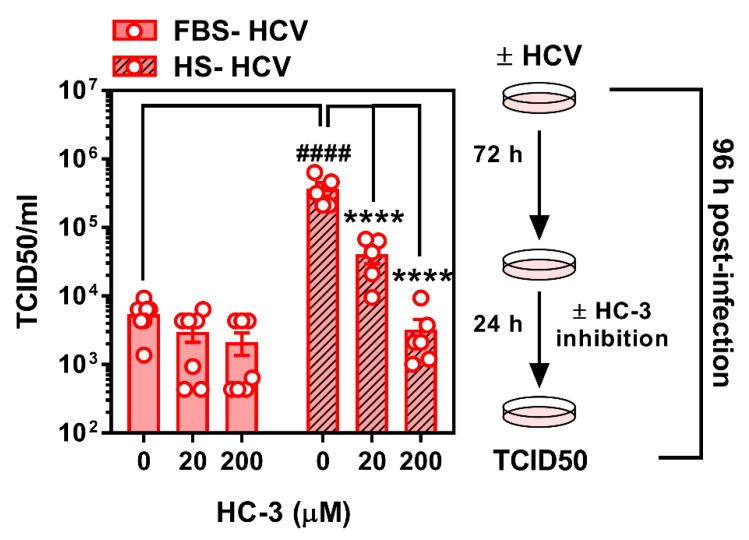
Inhibition of choline uptake decreases HCV infectivity. FBS- and HS-cultured Huh7.5 cells were infected at an MOI of 1 for 96 h in the presence or absence of 20 or 200 μM hemicholinium-3 (HC-3) to inhibit choline uptake for the final 24 h of infection. Media was removed to assess TCID50 as a measure of HCV infectivity. Data are mean ± SEM and are represent of 5–7 independent experiments. Statistical significance was determined by two-way ANOVA with a Tukey post-hoc analysis, such that **** represents *p* < 0.0001 compared to vehicle control within serum culture condition and #### represents *p* < 0.0001 compared between FBS- and HS-cultured conditions.

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
