# Peer review of "In Vitro Hepatitis C Virus Infection and Hepatic Choline Metabolism"

_viruses, 2020, doi:10.3390/v12010108_

Round 1

Reviewer 1 Report

Odai et al. investigated the interaction between choline metabolism and early HCV infection. They found that choline transport primarily through intermediate affinity transporter choline transporter-like family (CTL) in FBS-cultured Huh7.5 cells. CTL1 expression and incorporation of choline to phosphatidylcholine (PC) was decreased and inhibition of choline uptake by the pharmacological inhibitor (HC3) increased viral replication in FBS-cultured Huh7.5 cells. However, the link between choline metabolism and HCV infection seems to be different in human serum (HS)-cultured Huh7.5 cells. Overall, this study generated some interesting observations, but there are several issues need to be further clarified.

One major concern is the seemingly different choline metabolism and interaction with HCV infection in FBS and HS-cultured Huh7.5 cells. However, the difference between these two cell culture conditions cannot be affirmatively compared because of their different experimental designs. For example, in HS-cultured cells, choline metabolism (Fig 5A) and HCV replication in response to HC3 inhibition (Fig 5B) were examined at 96 hr post HCV infection. It is unclear why the authors chose the timing different from the experiment in FBS-cultured cells. The latter was done at 24 hr post HCV infection. For a fair comparison, the timing for these experiments should be the same in order to know the influence of FBS and HS on cultured Huh7.5 cell line. In Fig 4B, it is shown that a high dose of HC3 (200 micromolar) could increase the replication of HCV. However, there is always some concern about the off-target effect of pharmacological inhibitors. To address this issue, the authors should use siRNA to knock down the expression of CTL1 to see whether the increase of HCV replication is indeed due to the suppression of choline uptake. In Fig 5D, it is shown that the infectious ability of secreted virus was reduced by HC3 treatment although the intracellular HCV RNA did not change in HC3-treated HS-cultured Huh7.5 cells. This is an interesting observation. However, it will be also interesting to know whether HC3 treatment of HCV-infected FBS-cultured Huh7.5 cells will also cause different infectious ability of secreted virions. This should be examined.

Authors Responses:

Reviewer 1-
Odai et al. investigated the interaction between choline metabolism and early HCV infection. They found that choline transport primarily through intermediate affinity transporter choline transporter-like family (CTL) in FBS-cultured Huh7.5 cells. CTL1 expression and incorporation of choline to phosphatidylcholine (PC) was decreased and inhibition of choline uptake by the pharmacological inhibitor (HC3) increased viral replication in FBS-cultured Huh7.5 cells. However, the link between choline metabolism and HCV infection seems to be different in human serum (HS)-cultured Huh7.5 cells. Overall, this study generated some interesting observations, but there are several issues need to be further clarified.

One major concern is the seemingly different choline metabolism and interaction with HCV infection in FBS and HS-cultured Huh7.5 cells. However, the difference between these two cell culture conditions cannot be affirmatively compared because of their different experimental designs. For example, in HS-cultured cells, choline metabolism (Fig 5A) and HCV replication in response to HC3 inhibition (Fig 5B) were examined at 96 hr post HCV infection. It is unclear why the authors chose the timing different from the experiment in FBS-cultured cells. The latter was done at 24 hr post HCV infection. For a fair comparison, the timing for these experiments should be the same in order to know the influence of FBS and HS on cultured Huh7.5 cell line.

Response: We now show a direct comparison between the FBS and human serum (HS) culture system, both at 24 and 96 h post-HCV infection. There are inherent differences between FBS and HS-cultured Huh7.5 cells in terms of HCV infection that have previously been characterized. However, no study has interrogated potential differences in choline transporters and the importance of choline uptake in these contexts. We now demonstrate that the main choline transporters (CTL1, CTL2 and OCT1) show unique expression patterns between FBS- and HS-cultured cells and that HCV infection only has a negative effect on the transcript and protein expression of CTL1 (SLC44A1) at an early time point of infection in FBS-cultured cells. This is associated with diminished choline transport. However, by 96 h post-HCV infection, expression levels were corrected.

In addition, we now demonstrate that at 24 h post-HCV infection, inhibition of choline transport augments relative intracellular viral RNA in both FBS- and HS-cultured cells. However, at 96 h post-HCV infection, only the highest dose of choline transport inhibitor in HS-cultured cells had this effect. Importantly, and counter to this, TCID50 results demonstrate that choline uptake inhibition for the last 24 h of the 96 h infection dramatically lowers the number of infectious particles in HS- but not FBS-cultured cells. The mechanisms driving these effects, unfortunately, will have to be the focus of future manuscripts.

In Fig 4B, it is shown that a high dose of HC3 (200 micromolar) could increase the replication of HCV. However, there is always some concern about the off-target effect of pharmacological inhibitors. To address this issue, the authors should use siRNA to knock down the expression of CTL1 to see whether the increase of HCV replication is indeed due to the suppression of choline uptake.

Response: While siRNA knockdown of specific transporters would be an elegant way of showing individual transporter contribution to this phenotype, the logistical complexity of this was limiting. Huh-7-derived cells are hard to transfect, and transfection rates are never higher than 50-70% in FBS cultured cells, which may limit the potential interpretation. HS cultured cells, like primary hepatocytes are even harder to transfect, and therefore stable knockdowns must be generated in FBS, followed by differentiation. We expect that stable reduction of CTL1 (the only choline transporter whose expression is shown to be modified here) would have many other effects (a recent paper has just been published in Brain; PMID: 31855247, detailing the human phenotype of loss-of-function CTL1 mutations), which would again, complicate potential interpretation of the results. For these reasons we consider it not feasible to perform siRNA for this study.

We rationalize the use of the inhibitor as being able to inhibit all high- and intermediate-affinity choline transport at once and that the use of two concentrations might help tease out specific effects. However, the potential for off-target effects remains and is acknowledged as a limitation.

In Fig 5D, it is shown that the infectious ability of secreted virus was reduced by HC3 treatment although the intracellular HCV RNA did not change in HC3-treated HS-cultured Huh7.5 cells. This is an interesting observation. However, it will be also interesting to know whether HC3 treatment of HCV-infected FBS-cultured Huh7.5 cells will also cause different infectious ability of secreted virions. This should be examined.

Response: We agree and have now completed these experiments and describe them above for the reviewer.

Reviewer 2 Report

The manuscript by Odai et al analyses the hepatic choline metabolism upon HCV infection. In general, the story is well written and designed, the manuscript and the authors conclusion are comprehensible. However, some points should be addressed.

Major aspects:

An additional assay (TCID50) would help to validate the findings that cholin inhibition alters HCV life cycle progression in FCS-cultured Huh7.5 cells (as performed in Fig 5 for HS-cultured Huh7.5 cells).

For a reliable conclusion, experiments in Figure 5 should be performed simultaneously with FCS-cultured cells. Otherwise, a link between the experiments performed with HS and FCS is missing and it is hard to judge whether the observed effects are based on the altered cultivation or the different time point.

Minor aspects:

Line 107: Did the DMEM contain L-Glut and NEAA?

Line 211: It is mentioned that qunine does not affect choline uptake “at the highes dose”, but results are shown for only one concentration. Please rephrase the sentence. In addition, there are no information about the Quinine inhibition experiments in Material & Methods or in the main text.

Line 231 and legend of figure 2 and figure 3: Please clarify whether cells were infected at MOI 1 or MOI 2.

Legend of figure 2: Please specify that the cells are HCV infected.

Figure 3A: The relative expression of several of the tested genes is not 1.0, although it is normalized. Please revise it.

Line 256 and legend of figure 4: Please clarify whether cells were infected at MOI 1 or MOI 2.

Line 259: Please rephrase the sentence “… whereas 200 μM treatment resulted in a dramatic reduction…”, since its only a ~3-fold change.

Figure 4B: Please indicate the meaning of ###.

Figure 5D: There are 4 data points for 200 µM but only two for 0 µM and 20 µM. Are some data points missing for 0 µM and 20 µM?

In general, figure legends are very short. Providing more information would help to better understand the experiments and figures.

Authors Responses

The manuscript by Odai et al analyses the hepatic choline metabolism upon HCV infection. In general, the story is well written and designed, the manuscript and the authors conclusion are comprehensible. However, some points should be addressed.

Major aspects:
An additional assay (TCID50) would help to validate the findings that choline inhibition alters HCV life cycle progression in FCS-cultured Huh7.5 cells (as performed in Fig 5 for HS-cultured Huh7.5 cells). For a reliable conclusion, experiments in Figure 5 should be performed simultaneously with FCS-cultured cells. Otherwise, a link between the experiments performed with HS and FCS is missing and it is hard to judge whether the observed effects are based on the altered cultivation or the different time point.

Response: We agree and have now completed these experiments. We now demonstrate that at 24 h post-HCV infection, inhibition of choline transport augments relative intracellular viral RNA in both FBS- and HS-cultured cells. However, at 96 h post-HCV infection, only the highest dose of choline transport inhibitor in HS-cultured cells had this effect. Importantly, and counter to this, TCID50 results demonstrate that choline uptake inhibition for the last 24 h of the 96 h infection dramatically lowers the number of infectious particles in HS- but not FBS-cultured cells. The mechanisms driving these effects, unfortunately, will have to be the focus of future manuscripts.

Minor aspects:
Line 107: Did the DMEM contain L-Glut and NEAA?

The culture media (DMEM) did contain L-Glutamine (2 mM) but not NEAA. These conditions were modeled from previous publications.

Line 211: It is mentioned that quinine does not affect choline uptake “at the highest dose”, but results are shown for only one concentration. Please rephrase the sentence. In addition, there are no information about the Quinine inhibition experiments in Material & Methods or in the main text.

We apologize for this mistake and have now corrected the wording regarding this experiments. Additionally, we have included a methodological description in the materials and methods.

Line 231 and legend of figure 2 and figure 3: Please clarify whether cells were infected at MOI 1 or MOI 2.

All infection experiments were completed with an MOI of 1. This is now clarified/standardized in the main text.

Legend of figure 2: Please specify that the cells are HCV infected.

This has been specified.

Figure 3A: The relative expression of several of the tested genes is not 1.0, although it is normalized. Please revise it.

This has been addressed in the revised manuscript.

Line 256 and legend of figure 4: Please clarify whether cells were infected at MOI 1 or MOI 2.

All infection experiments were completed with an MOI of 1. This is now clarified/standardized in the main text.

Line 259: Please rephrase the sentence “… whereas 200 μM treatment resulted in a dramatic reduction…”, since its only a ~3-fold change.

This has been rephrased to de-emphasize this (now revised) result.

Figure 4B: Please indicate the meaning of ###.

This has been corrected.

Figure 5D: There are 4 data points for 200 μM but only two for 0 μM and 20 μM. Are some data points missing for 0 μM and 20 μM?

To clarify this point, we have conducted new experiments and display the results of 3-5 biological replicates for most figures and 5-7 for figure 6.

In general, figure legends are very short. Providing more information would help to better understand the experiments and figures.

We have added information to the figure legends that we think make them easier to understand. Thank you for the suggestion.

Round 2

Reviewer 1 Report

All issues addressed.

Reviewer 2 Report

The authors have addressed my concerns satisfactorily.